# Genetic variation of the HIV-1 subtype C transmitted/founder viruses long terminal repeat elements and the impact on transcription activation potential and clinical disease outcomes

**Paradise Madlala**[1,2]*, **Zakithi Mkhize**[1], **Shamara Naicker**[1], **Samukelisiwe P. Khathi**[1], **Shreyal Maikoo**[1], **Kasmira Gopee**[1], **Krista L. Dong**[3,4,5], **Thumbi Ndung'u**[1,2,3,6,7]

**1** HIV Pathogenesis Programme, The Doris Duke Medical Research Institute, Nelson R. Mandela School of Medicine, University of KwaZulu-Natal, Durban, South Africa, **2** School of Laboratory Medicine and Medical Sciences, University of KwaZulu-Natal, Durban, South Africa, **3** Ragon Institute of Massachusetts General Hospital, Massachusetts Institute of Technology, and Harvard University, Cambridge, Massachusetts, United States of America, **4** Division of Infectious Diseases, Massachusetts General Hospital, Boston, Massachusetts, United States of America, **5** Harvard Medical School, Boston, Massachusetts, United States of America, **6** Africa Health Research Institute (AHRI), Durban, South Africa, **7** Division of Infection and Immunity, University College London, London, United Kingdom

* madlalap@ukzn.ac.za.

## Abstract

A genetic bottleneck is a hallmark of HIV-1 transmission such that only very few viral strains, termed transmitted/founder (T/F) variants establish infection in a newly infected host. Phenotypic characteristics of these variants may determine the subsequent course of disease. The HIV-1 5' long terminal repeat (LTR) promoter drives viral gene transcription and is genetically identical to the 3' LTR. We hypothesized that HIV-1 subtype C (HIV-1C) T/F virus LTR genetic variation is a determinant of transcriptional activation potential and clinical disease outcome. The 3'LTR was amplified from plasma samples of 41 study participants acutely infected with HIV-1C (Fiebig stages I and V/VI). Paired longitudinal samples were also available at one year post-infection for 31 of the 41 participants. 3' LTR amplicons were cloned into a pGL3-basic luciferase expression vector, and transfected alone or together with Transactivator of transcription (*tat*) into Jurkat cells in the absence or presence of cell activators (TNF-α, PMA, Prostratin and SAHA). Inter-patient T/F LTR sequence diversity was 5.7% (Renge: 2–12) with subsequent intrahost viral evolution observed in 48.4% of the participants analyzed at 12 months post-infection. T/F LTR variants exhibited differential basal transcriptional activity, with significantly higher Tat-mediated transcriptional activity compared to basal (p<0.001). Basal and Tat-mediated T/F LTR transcriptional activity showed significant positive correlation with contemporaneous viral loads and negative correlation with CD4 T cell counts (p<0.05) during acute infection respectively. Furthermore, Tat-mediated T/F LTR transcriptional activity significantly correlated positively with viral load set point and viral load; and negatively with CD4 T cell counts at one year post infection (all p<0.05). Lastly, PMA, Prostratin, TNF-α and SAHA cell stimulation resulted in enhanced yet

**Data Availability Statement:** All relevant data are within the paper and its Supporting Information files.

**Funding:** Research reported in this publication was supported by the South African Medical Research Council with funds received from the National Department of Health (MRC-RFA-SHIP 02-2018 to PM) and a grant from the National Research Foundation Thuthuka Funding Instrument (TTK160529166617 to PM). This work was also supported in part by grants from the Bill and Melinda Gates Foundation (grant numbers OPP1212883 to TN and INV-033558 to TN), the International AIDS Vaccine Initiative (IAVI) (UKZNRSA1001 to TN) and the South African Research Chairs Initiative (grant # 64809 to TN). Additional funding was received from the Sub-Saharan African Network for TB/HIV Research Excellence (SANTHE), a DELTAS Africa Initiative (grant # DEL-15-006 to TN). The Developing Excellence in Leadership, Training and Science in Africa (DELTAS Africa) programme is supported by the Wellcome Trust (grant # 107752/Z/15/Z to TN) and the UK Foreign, Commonwealth & Development Office. The views expressed in this publication are those of the author(s) and not necessarily those of the Wellcome Trust or the UK government. JKM was supported by the FLAIR Fellowship Programme, which is a partnership between the African Academy of Sciences and the Royal Society funded by the UK Government's Global Challenges Research Fund (FLR\R1\201494 to TN). For the purpose of open access, the author has applied a CC BY public copyright licence to any Author Accepted Manuscript version arising from this submission. The funders had no role in study design, data collection and analysis, decision to publish, or preparation of the manuscript.

**Competing interests:** The authors have declared that no competing interests exist.

heterologous transcriptional activation of different T/F LTR variants. Our data suggest that T/F LTR variants may influence viral transcriptional activity, disease outcomes and sensitivity to cell activation, with potential implications for therapeutic interventions.

## Author summary

There is heterogeneity in the rates of clinical disease progression in antiretroviral therapy-naïve people living with HIV (PLWH). In heterosexual HIV-1 transmission, only a single or very few viral strains, called transmitted/founder (T/F) viruses establish infection in a newly infected host. The long terminal repeat (LTR) is the viral promoter that drives viral gene transcription and is important for the HIV-1 life cycle. In this study we investigated the impact of HIV-1 subtype C T/F virus LTR genetic variation on transcriptional activity, clinical disease outcomes and response to cell activation. Our data show inter-patient T/F LTR genetic variation and limited intrahost evolution by 12 months post infection. T/F LTR variants exhibit differential basal LTR transcriptional activity, which is significantly increased in the presence of the Transactivator of transcription (Tat) protein. Furthermore, we show that T/F LTR transcription activity significantly correlates positively with viral load and viral load set point but negatively with CD4 T cell count. Lastly, we show that T/F LTR variants exhibit differential responses to cell activators PMA, TNF-α, Prostratin and SAHA. Taken together our data suggest that T/F viruses LTR genetic variation and functional heterogeneity are important determinants of clinical outcomes and virus reactivation potential.

## Introduction

A hallmark of untreated human immunodeficiency virus type 1 (HIV-1) infection is heterogeneity in clinical disease progression rates [1,2]. It is well known that there are remarkable differences in peak viremia immediately following HIV-1 infection, and in the viral load set point thereafter, with the later parameter known to be a prognostic indicator of the rate of disease progression [2]. Transmitted/founder virus genetic variants and their associated phenotypic characteristics such as the virus replication capacity and biological functions of specific genes may contribute to differences in the rate of disease progression [3–7]. However, the full extent of viral factors that impact disease progression is not fully understood, which hampers the development of novel strategies to prevent and treat HIV-1 infection. Nevertheless, effective combination antiretroviral therapies (cART) have been developed that can effectively suppress plasma HIV-1 RNA, although these therapies are not curative [8]. The failure of cART to eradicate HIV-1 is due to the establishment of a latent viral reservoir immediately following infection, which then leads to almost inevitable viral rebound when treatment is interrupted [9]. The viable latent reservoir consists of cells that are infected with replication competent and yet transcriptionally silent proviruses [9,10]. HIV-1 latency is seeded and maintained through genetic and epigenetic mechanisms that create specific repressive chromatin configuration at the 5' long terminal repeat (LTR) viral promoter [11,12]. The 5' LTR, which is identical to the 3' LTR drives HIV-1 gene transcription and is important for the viral life cycle while 3' LTR is involved in gene transcription termination [13]. The LTR is divided into the U3, R and U5 regions. The U3 region contains three functional domains (modulatory, core enhancer and core promoter) that regulate HIV-1 positive sense transcription (reviewed in [14]). Each of

these regions interact with an array of viral and host factors to mediate viral gene transcription and virus replication.

In approximately 80–90% of heterosexual infections, due to a transmission bottleneck, HIV-1 infection is established by a single, or few genetic variants named the transmitted/founder (T/F) virus [3]. This transmission bottleneck selects for viral strains that more readily cross the mucosal barrier, productively infect target cells and establish systemic infection [15,16]. T/F viruses exhibit CCR5 coreceptor usage with shortened Envelope variable loops [17] and enhanced resistance to interferon alpha (IFN-$\alpha$) [18–20]. Other studies showed that heterosexual transmission selects for T/F viruses that have subtype consensus-like sequences [3,18,21], which in some studies have been associated with lower virus replication capacity [18,21,22]. However, the data are conflicting with other studies suggesting the preferential transmission of viruses with high replication capacity, or context dependent transmission of particular variants [3,23].

It has been reported that HIV-1 subtype C (HIV-1C) possesses 3 to 4 NF-$k$B binding sites in the LTR, unlike other subtypes that contain only 1 or 2 of these sites, with consequent differences in viral transcriptional activity [24–27]. A study from India reported that variants containing 4 NF-$k$B sites are replacing the standard HIV-1C viruses that exhibit 3 NF-$k$B binding sites [28]. Furthermore, these 4 NF-$k$B binding sites containing viruses were associated with significantly higher viral loads in people living with HIV-1 (PLWH) [28]. Taken together these data suggest that the emerging 4 NF-$k$B viral strains probably have a selective transmission advantage and are more pathogenic. However, consistent with a previous report [29], a recent study [30] showed that only 6% of HIV-1C viruses circulating in South Africa exhibit the fourth NF-$k$B binding site, suggesting that these 4 NF-$k$B viral strains do not have selective advantage in epidemic spread. All of the aforementioned studies investigated LTR genetic variation during chronic infection where diverse quasispecies are typically present. Moreover, genetic variation may extend to other domains and regions within the LTR elements, and few studies have investigated the impact of this variation in the LTR on virus functional characteristics and clinical disease outcomes.

Despite its central role in driving viral gene transcription and importance in viral life cycle, the genetic variation of the LTR element in T/F viruses and its impact on disease outcome and transcriptional activation potential has not been extensively characterized. Understanding the impact of the LTR genetic variation on disease outcome and viral transcriptional activity may guide the development of novel HIV therapeutic or cure strategies. Therefore, the major aim of this study was to characterize the genetic and functional heterogeneity of HIV-1C T/F LTRs, and to correlate this with markers of disease progression in South African individuals followed longitudinally after detection of acute HIV-1 infection. We also assessed the transcriptional activity of HIV-1C T/F LTR variants following activation with diverse stimulants, including proposed latency reversal agents. We describe inter-patient HIV-1C T/F LTR genetic variation and intra-patient evolution by 12 months post infection. Our data show that HIV-1C T/F LTR exhibited differential transcriptional activity, which is enhanced in the presence of Tat. We demonstrate significant positive correlation between HIV-1C T/F LTR transcriptional activity with viral load and viral load set point but negative correlation with CD4 T cell counts. Lastly, we show a heterogeneous HIV-1C T/F LTR transcriptional activity response following stimulation with diverse potential latency reversal agents.

## Results

### Characteristics of people living with HIV-1 (PLWH)

The demographic and clinical characteristics of the 41 participants from the FRESH and HPP AI cohorts (Fig 1) included in this study are shown in Table 1. The average age at virus detection within the FRESH and HPP AI cohorts was 23 years (± SD, 1.80) and 32 years (±SD, 11.40) respectively. All 21 participants from the FRESH cohort were female (100%), compared to 13 of 20 (65%) from the HPP AI cohort. Plasma samples were obtained at an early timepoint within a median of 1.00 day (IQR, 0.30–3.00) of virus detection from 21 patients (Fiebig I) with median square root CD4 T cell count of 27.30 (IQR, 21.15–30.18) or median CD4 T cell count of 745.54 (IQR, 447.69–910.53) (S1 Table) and plasma viral load of $\log_{10}$ 4.80 copies/mL (IQR, 3.80–5.45). The earliest available samples from the rest of 20 participants (Fiebig V/VI) were obtained at a median of 34 days (IQR, 30.00–40.75) post infection with the median square root CD4 T cell count of 21.30 (IQR, 19.55–25.83) or median CD4 T cell count of 453.70 (IQR, 283.39–616.37) (S1 Table) and viral load of $\log_{10}$ 5.10 copies/mL (IQR, 3.90–5.78). While the median absolute CD4 T cell count was significantly different between the group whose samples were available within a median of 1 day of virus detection and 34 days post infection at the earliest time point (p = 0.0240), the median square root CD4 T cell count is reported in this study because a square root transformation has the effect of making the data less skewed and variation more uniform. The median rate of CD4 T cell decline per month amongst 14 ART naïve FRESH participants (following acute infection to 12 moths post infection) calculated by linear regression was −15.90 cells/mm$^3$ (IQR, −27.42 to -2.67). The median viral load set point (from 3 to 12 months post infection) was $\log_{10}$ 4.80 copies/mL (IQR, 3.90–5.05). In the HPP AI cohort the median rate of CD4 T cell decline per month up to one year post the first positive HIV-1 RNA calculated by linear regression was −6.79 cells/mm$^3$ (IQR, −12.01 to − 0.20) and the median $\log_{10}$ viral load set point was 4.70 copies/mL (IQR, 4.10–5.10). Matching plasma samples at one-year post infection were obtained from 31 of 41 patients. Therefore, a total of 72 plasma samples, 41 obtained at earliest timepoint and 31 obtained at approximately one-year post infection timepoint were available for this study.

### Characterization of HIV-1C long terminal repeat (LTR) genetic variation

We hypothesized that HIV-1C T/F LTR sequence genetic variation may affect transcriptional activity and disease outcome. To evaluate this hypothesis, a total of 72 LTR sequences, 41 obtained at or immediately after the acute phase of infection (referred to as acute infection timepoint henceforth) and 31 obtained at one-year post infection were generated and used for phylogenetic analysis. Specifically, 41 LTR sequences obtained at acute infection timepoint were used to study genetic variation at transmission and matched 31 LTR sequences obtained one year later were used to assess diversification following acute HIV-1 infection.

Phylogenetic analysis demonstrated that the LTR sequences from individual participants were unlinked and belonged to HIV-1C as they rooted to HIV-1C LTR consensus sequence (Fig 2). The phylogenetic tree shows intra-patient clustering of LTR nucleotide sequences obtained at acute and one-year timepoints and that LTR sequences generated at the acute timepoint, without matching later timepoint sample, were distinct and did not form clusters. Furthermore, our data demonstrate that there was a median of 45 (IQR: 39–53) inter-patient T/F LTR sequence diversity with subsequent intrahost viral evolution observed in 48.4% of the participants analyzed at 12 months post-infection. The median pairwise diversity between the early (acute) and late timepoints was 2 (IQR: 0–4) with 61% of the sequences exhibiting little to no inter-timepoint diversity (i.e. SNPs distance between 0–2).

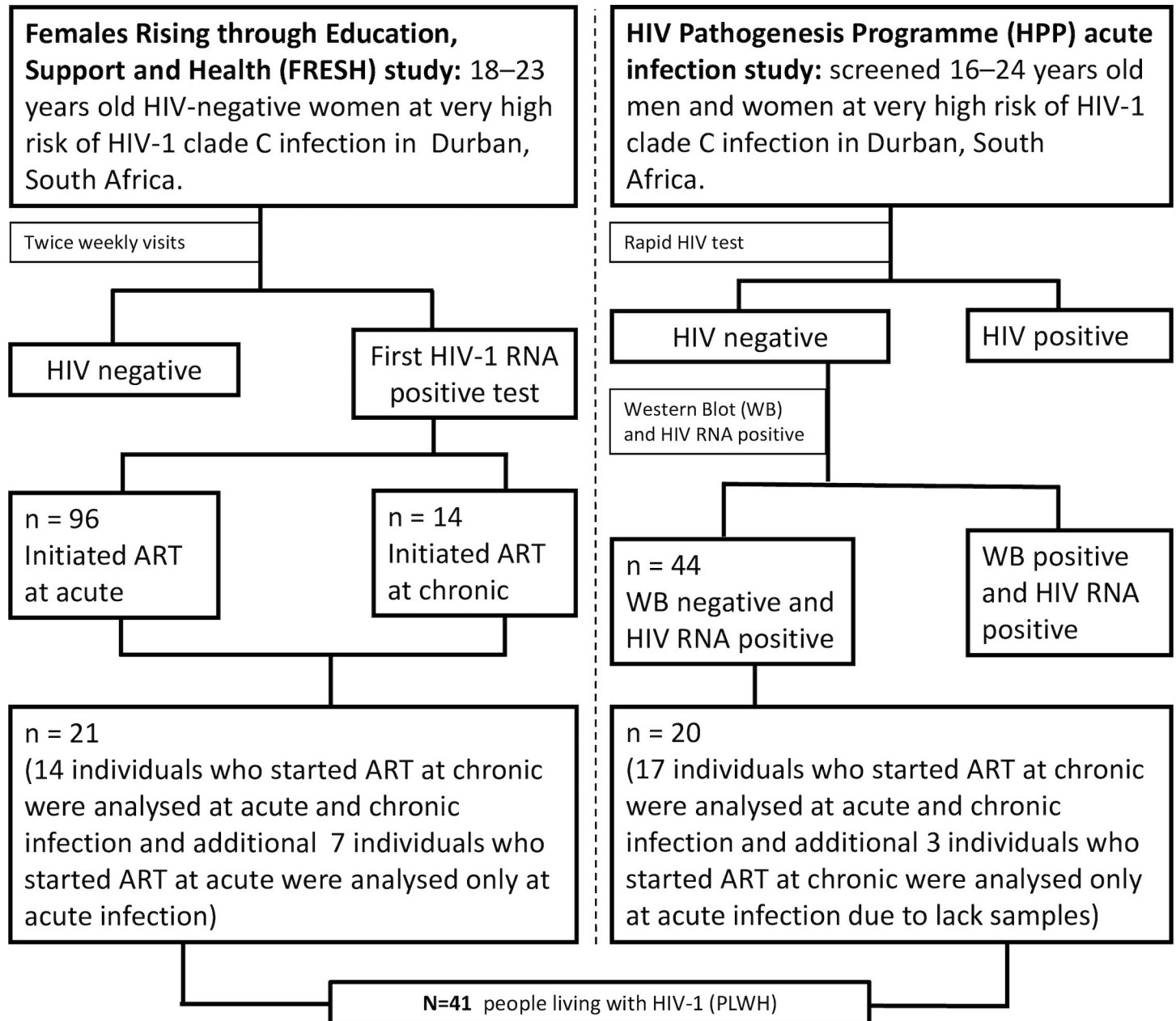

**Fig 1. Design of South African acute HIV-1 infection cohorts used in this study and the number of participants from each cohort used to generate HIV-1 subtype C LTR sequences.** Initiated treatment immediately refers to the study participants that initiated treatment immediately after detection of HIV RNA. Initiated treatment late refers to the study participants who initiated treatment when their CD4 T cell count was below 500 cells/µL as was the South African national treatment guidelines at the time.

Characterization of HIV-1 LTR sequence genetic variation at acute and evolution by one-year post infection timepoints may guide the development of novel strategies to prevent and treat HIV-1 infection. We explored whether branch length differences in the phylogenetic tree were due to variation within HIV-1C LTR transcription factor binding sites (TFBS) and diversification during primary infection. HIV-1C LTR sequences obtained at the acute infection timepoint were compared to matched LTR sequences obtained at one-year post infection, aligned against an Indian subtype C consensus sequence previously reported to have 4 NF-$k$B

**Table 1. The demographic and clinical characteristics of the 41 people living with HIV-1 used in this study.**

| Characteristics | At viral RNA detection or enrolment | | | At one-year post infection | | |
|---|---|---|---|---|---|---|
| | FRESH | HPP Cohort | p-value | FRESH | HPP cohort | p-value |
| Number of patients (%) | 21 (51.22) | 20 (48.78) | 0.8714 | 14 (45.16) | 17 (54.84) | 1.0000 |
| Age (yrs) mean ± SD (range) | 21.20 ± 1.80 | 32.80 ± 11.40 | 0.2016 | N/A | N/A | N/A |
| Gender, female (%) | 21/21 (100.00) | 13/20 (65.00) | 0.4838 | 14/14 (100.00) | 13/17 (76.47) | 0.7926 |
| Fiebig Stage (number of patients) | I (15) | V/VI (20) | N/A | I (14) | V/VI (17) | N/A |
| Matched samples obtained at early timepoint/late timepoint (%) | 14/21 (66.67) | 17/20 (85.00) | 0.6106 | N/A | N/A | N/A |
| Median days post virus detection or enrolment (IQR) | 1 (IQR, 0.25–3.00) | 34 (IQR, 30.00–40.75) | 0.0001 | N/A | N/A | N/A |
| Median square root CD4 T cell counts (IQR) | 27.30 (IQR, 21.15–30.18) | 21.30 (IQR, 19.55–24.83) | 0.6351 | 22.14 (IQR, 16.39–24.17) | 20.02 (IQR, 16.97–21.86) | 0.3179 |
| Median Log$_{10}$ HIV RNA (IQR) | 4.80 (IQR, 3.80–5.45) | 5.10 (IQR, 3.90–5.78) | 0.9448 | 3.83 (IQR, 3.37–5.04) | 3.83 (IQR, 4.05–4.88) | 0.8339 |
| Median rate of CD4 T cell decline per month (IQR) | −15.90 (IQR,−27.42-−2.67) | −6.79 (IQR,−12.01-−0.20) | 0.5855 | N/A | N/A | N/A |
| Median Log$_{10}$ viral load set point (IQR) | 4.80 (IQR, 3.90–5.05) | 4.70 (IQR, 410–510) | 0.9448 | N/A | N/A | N/A |

- N/A means the p-value could not be calculated or the infomation at viral RNA detection or enrolment timepoint is the same as the information at one-year post information.

- IQR (interquartile range).

sites [28] instead of the 3 NF-*k*B sites found in the standard HIV-1C viruses (Fig 3A). Previous reports indicated that consensus-like sequences are preferentially transmitted and therefore dominate in acute HIV-1 infection [31,32]. We therefore assessed the percentage (%) of similarity of each LTR sequence to the consensus for both early and late timepoints and found there was no difference in median % similarity between the two timepoints (Fig 3B). Our data demonstrate that LTR sequences generated from acute and one-year post infection timepoints were generally similar within an individual, comprises the U3 (Fig 3A) and R (S2 Fig) regions which is typical of the LTR element of the HIV-1 RNA genome as previously reported [25]. The R region contains the Trans-activation response (TAR) RNA element. The U3 region is composed of core-promoter, core-enhancer and modulatory domains. Specifically, the core-promoter exhibits 2 E-box motifs, TATA box and 3 Sp1 (I, II, III) binding sites while the core-enhancer domain exhibits at least 3 NF-*k*B binding sites for subtype C infection.

Our data show that the TAR sequence is polymorphic with 41/41 (100%) of the T/F LTR sequences exhibiting nucleotide variation at least one position compared to the reference sequence. A total of 5/41 (8.1%) T/F LTR sequences exhibited genetic variation within the 3' E-box (5'-CAGCTG-3' ➔ 5'-CCGCCG-3') while the other 3/41 (7.3%) T/F LTR sequence exhibited variation within the 5' E-box (5'-CAGATG-3' ➔ 5'-CAAAAG-3'). Only 2/41 (4.9%) T/F LTR sequences exhibited TATA box variation (5'-TATAA-3' ➔ 5'-TAAAA-3'), which is more characteristic of HIV-1 subtype E viruses [24]. Next we show that 31/41 (75.6%) T/F LTR sequences exhibited polymorphism within the Sp1I binding site (5'-GGGAGTGGC-3' ➔ 5'-GGGAGTGGT-3'). The Sp1II binding site was variable in 5/41 (12.2%) T/F LTR sequences exhibiting (5'-GGGCGGGAC-3' ➔ 5'-GGGCGGGAA-3' or 5'-GGGCGGGCT-3' or 5'-GGGCGGTAC-3' or 5'-GGTCGGGAC-3') variation. Lastly, the Sp1III binding site was also polymorphic, with the Sp1III_G2A variant (5'-GGGGAGTGGT-3 ➔ 5'-GAGGAGTGGT-3') noted in 24/41 (58.5%) T/F LTR sequences while 24/41 (58.5%) T/F LTR sequences exhibited the (5'-GGGGAGTGGT-3 ➔ 5'-GGGGTGTGGT-3') variation referred to as Sp1III_A5T variant and 24/41 (58.5%) T/F

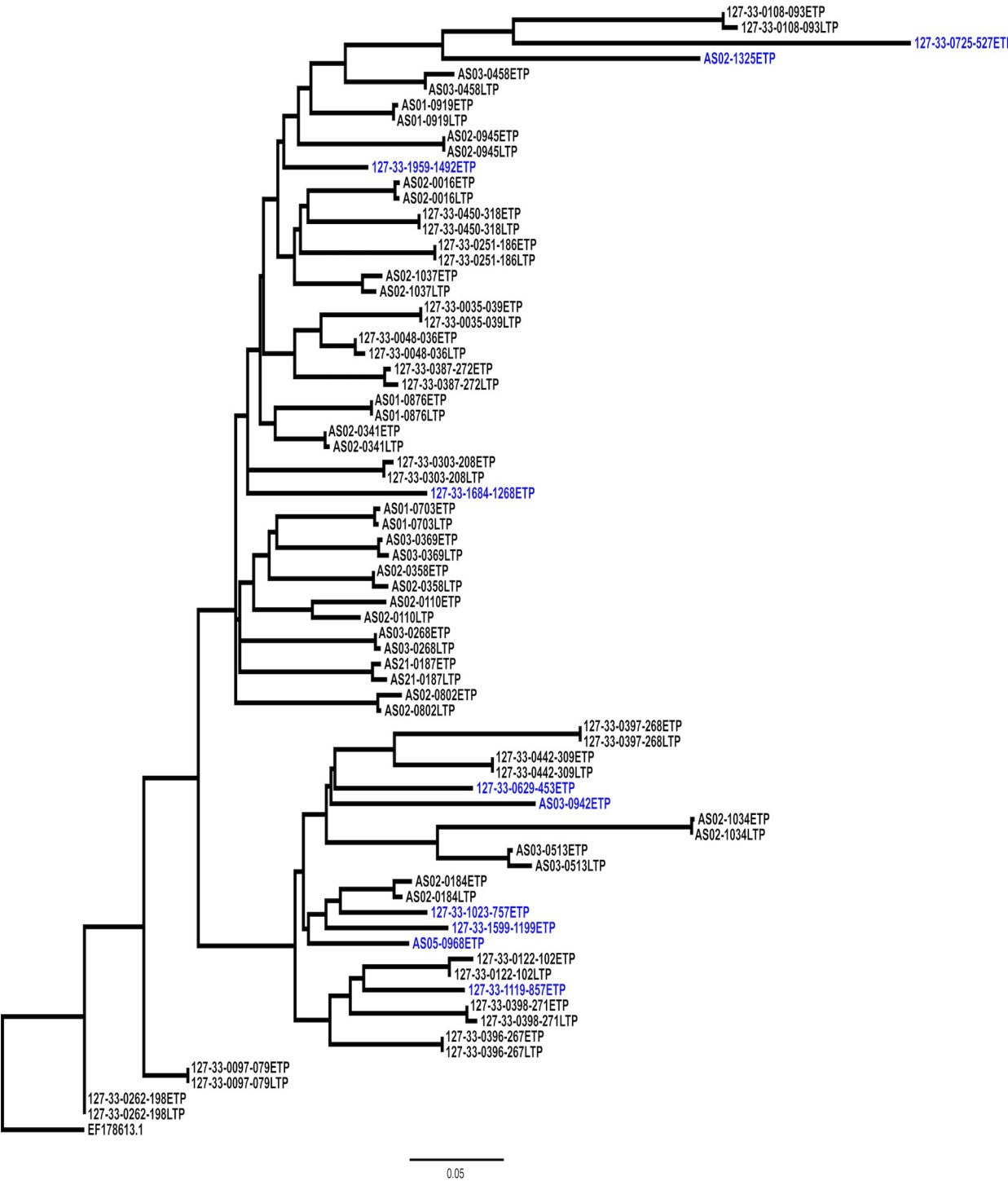

**Fig 2. Phylogenetic analysis of patient derived LTR elements obtained at the earliest timepoint (ETP) and one-year post infection later timepoint (LTP) from the FRESH and HPP cohort and transcriptional analysis of the transmitted/founder (T/F) LTR sequences. A**: The phylogenetic tree was constructed using the online tool Phyml (http://www.hivlanl.gov) and rerooted on the Indian subtype C reference sequence (EF178613.1) using Figtree Tree Figure Drawing Tool v1.4.3. The viral sequences obtained at or near transmission are denoted as (ETP) while the viral sequences obtained at one-year post infection are denoted as (LTP). The LTR sequences highlighted in blue did not have the matching sample at the LTP timepoint. Each patients LTR element clusters independently together thus demonstrating that the LTR element is variable between patients. Phylogenetic branching demonstrates that the LTR may evolve within one-year of infection within a patient.

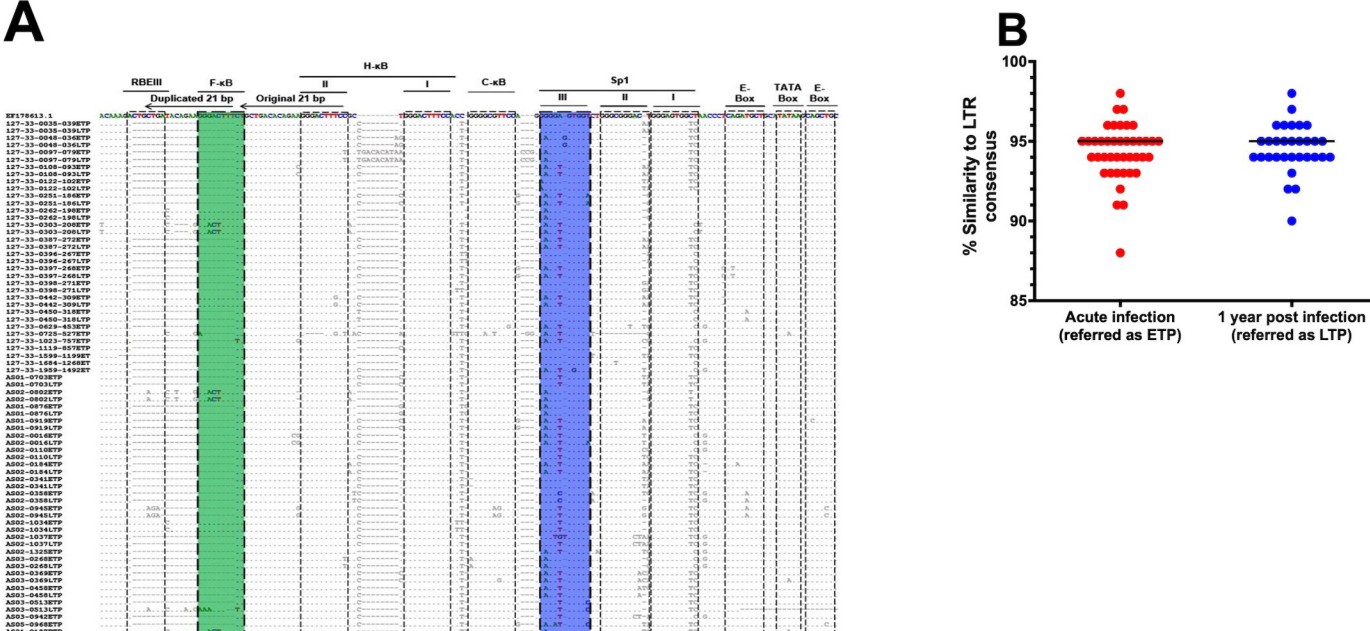

**Fig 3. Multiple sequence alignment of patient derived LTR sequences and percentage of similarity between patient derived and consensus LTR sequences. A:** The patient derived LTR sequences were aligned against the Indian subtype C reference sequence, EF178613.1 since it contains 4 NF-*k*B binding sites. The dots represent nucleotide bases that are identical to the reference sequence while the dashes indicate deletions. The doted blocks highlight the transcription factor binding sites (TFBS) within the core-enhancer and core-promoter regions and TAR element. From left to right: the USF RBE III site, the 4th NF-*k*B binding site (designated F-*k*B and highlighted in green), standard NF-*k*B sites designated NF-*k*B I and II (H- *k*B, II and I) and subtype C specific NF-*k*B site (designated C-*k*B), Sp1 III (highlighted in blue), II and I binding sites, 5' E-box, Tata box and 3' E-box and the TAR loop region. Overall, the T/F virus canonical sequences were relatively conserved within the core enhancer, nonetheless variation was observed within the RBE III site, Sp1 III binding site and TATA Box. **B:** Shows the percentage of similarity of each individual patients derived LTR at earlyt time point (ETP, shown in blue) and late time point (LTP, denoted in red) to the consensus sequence.

LTR sequences exhibited a double (5'-GGGGAGTGGT-3 ➔ 5'-G<u>A</u>GG<u>T</u>GTGGT-3) variation referred to as Sp1III_G2A/A5T variant [33].

Consistent with previous reports on HIV-1C viruses obtained during chronic phase of infection in South Africa [29,30], our data show that in addition to 3 types of NF-*k*B binding sites, C-*k*B (5'-GGGGCGTTCC-3'), H-*k*B [I (5'-GGGACTTTCC-3') and II (5'-GGGACTTTCC-3')] there is an insertion of the fourth NF-*k*B binding site, F-*k*B (5'-GGGACTTTCT-3'). A total of 4/41 (9.8%) of the T/F LTR sequences exhibited variation in the C-*k*B site (5'-GGGGCGTTCC-3' ➔ 5'-AGGGCGTTCC-3' or 5'-GGG<u>A</u>CTTTCC-3' or 5'- GGGGC<u>AG</u>TCC- 3'). All 41/41 (100%) T/F LTR sequences exhibited an intact H-*k*B site [I (5'-GGGACTTTCC-3')], 4/41 (9.8%) H-*k*B II site exhibited variation (5'-GGGACTTTCC-3' ➔ 5'-GGGACTT<u>G</u>CC-3' or G—-TT<u>GCT</u> or 5'-GGGACTTT<u>CT</u>-3', which is the F-*k*B site sequence). A total of 6/41 (14.6%) T/F LTR sequences exhibited F-*k*B site at its unique position as previously described [28], while only 2/41 (4.9%) exhibited the F-*k*B site sequence in place of the H-*k*B II site as previously described [30]. Three of 6 (50%) T/F LTR sequences exhibiting the F-*k*B site in its unique position exhibited an intact F-*k*B canonical sequence (5'-GGGACTTTCT-3'), while the remaining 3/6 F-*k*B sites exhibited variation (5'-GGGACTTTCT-3' ➔ 5'-GG<u>A</u>CTTTTCT-3'). Consistent with a previous study [34], our data further showed that these 6/41 (14.6%) T/F LTR sequences exhibiting the F-*k*B site contain an insertion of the RBEIII site sequence (5'-ACTGCTGA-3'). Taken together our data suggest that while H-*k*B II site canonical sequence remain conserved, extensive genetic

variation occurs across the U3 and R regions of the T/F LTR sequences and there is no apparent enrichment of viruses with four NF-$k$B sites among recently transmitted viruses.

## The impact of T/F LTR sequence variation on transcriptional activity and disease progression

Earlier studies reported that inter- and intra-subtype LTR genetic variation during chronic infection results in differential LTR-driven gene transcription [25,35]. However, the effect of HIV-1C T/F LTR genetic variation on transcriptional activity has not been fully studied. We investigated whether HIV-1C T/F LTR sequence variation results in differential ability to drive the expression (transcription) of a reporter *luciferase* gene. The ability of the 41 HIV-1C T/F LTR variants to drive basal and Tat-mediated transcription of the *luciferase* gene was assessed. The HIV-1C consensus or T/F LTR sequences cloned upstream of the *luciferase* gene present in the pGL3 Basic vector individually were transfected either alone or co-transfected together with recombinant pTarget plasmid containing either HIV-1C consensus or autologous Tat into Jurkat cell lines in order to assess their transcription activity. The data show that T/F LTR variants result in differential transcriptional activity, with the presence of Tat resulting in significantly higher (p<0.0001) transcriptional activity compared to basal LTR transcriptional activity, while consensus Tat-mediated transcriptional activity was significantly higher than that of autologous Tat-mediated T/F LTR (p = 0.0008) (Fig 4A).

Next, we hypothesized that HIV-1C T/F LTR transcriptional activity may be associated with markers of disease outcome. Basal (r = 0.4999, p = 0.0004), consensus Tat- (r = 0.4499, p = 0.0016) and autologous Tat- (r = 0.2940, p = 0.0310) mediated T/F LTR transcriptional activity showed significant positive correlation with viral load (Fig 4B-4D) but negative correlation (r = -0.3428, p = 0.0141; r = -0.3284, p = 0.0180 and r = -0.3514, p = 0.0121 respectively) with CD4 T cell count (S3A-S3C Fig) at the acute phase of infection. While basal T/F LTR transcriptional activity showed no correlation (r = -0.0512, p = 0.3922) with viral load (S3D Fig), consensus and autologous Tat-mediated T/F LTR transcriptional activity showed significant positive correlation (r = 0.4698, p = 0.0038 and r = 0.4052, p = 0.0119 respectively) with viral load (Fig 4E-4F) at one-year post infection. Consistently, there was no correlation (r = 0.1683, p = 0.1827) between basal T/F LTR transcriptional activity and CD4 T cell count (S3E Fig), while consensus and autologous Tat-mediated T/F LTR transcription activity exhibited significant negative correlation (r = -0.3341, p = 0.0331 and r = -0.4413, p = 0.0065 respectively) with CD4 T cell count (S3F-S3G Fig) at one year post infection.

To assess the impact of the T/F LTR genetic variation on disease outcome, the correlation between T/F LTR transcription activity and viral load set point (mean viral load from 3 to 12 months post infection) was assessed. While no correlation (r = -0.1100, p = 0.2745) between basal T/F LTR transcription activity and viral load set point (S3H Fig) was seen, the analysis revealed a significant positive correlation (r = 0.3576, p = 0.0223) between consensus Tat-mediated T/F LTR transcription activity and viral load set point (Fig 4G). Consistently, the data showed a significant positive correlation (r = 0.3382, p = 0.0292) between autologous Tat-mediated T/F LTR transcription activity and viral load set point (Fig 4H). However, there were no differences in transcriptional activity and viral loads between the T/F LTR sequences exhibiting 4 NF-$k$B versus 3 NF-$k$B sites (S4A Fig). Similarly, there were no differences in viral loads (S4B Fig) and CD4 T cell counts (S4C Fig) of participants infected with viruses exhibiting 4 NF-$k$B versus 3 NF-$k$B T/F LTR sequences. Transcriptional activity of the T/F LTR sequences exhibiting Sp1III_2G was comparable with T/F LTR varints exhibiting Sp1III_2A variation (S4D Fig). Viral loads and CD4 T cell counts of participants infected with T/F variants exhibiting Sp1III 2G variation were also comparable with those infected with Sp1III 2 viral variants

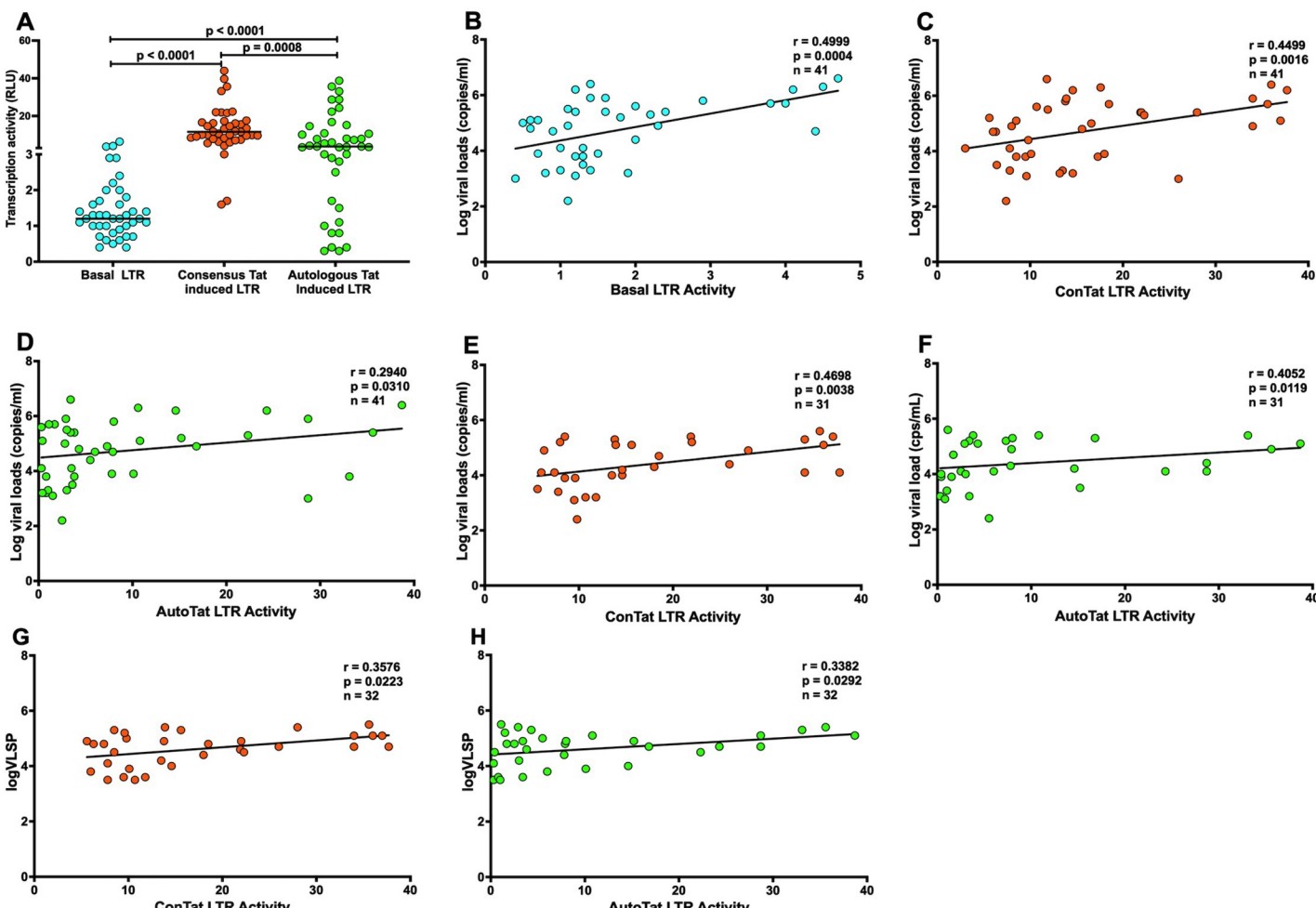

**Fig 4. T/F LTR transcriptional activity correlate with viral load. A**: Effect of T/F LTR sequence genetic variation on basal (shown in Cyan), consensus (shown in orange) and autologous (shown in green) Tat-mediated transcriptional activity. Recombinant consensus- or T/F LTR-pGL3 Basic vectors were transfected into Jurkat cells either alone or co-transfected with consensus or autologous Tat-pTarget recombinant plasmid and the luciferase activity was measured after a 24-hour culture period. The transfection assay for each sample was performed in triplicates therefore the transfection data presented here are illustrative of the average relative light units (RLU). **B:** Basal T/F LTR transcriptional activity exhibit significant positive correlation with viral loads at or near transmission (shown in Cyan). **C:** Consensus Tat-mediated T/F LTR transcriptional activity exhibit significant positive correlation with viral loads at or near transmission (shown in orange). **D:** Autologous Tat-mediated T/F LTR transcriptional activity exhibit significant positive correlation with viral loads at or near transmission (shown in green) **E:** Consensus Tat-mediated T/F LTR transcriptional activity exhibit significant positive correlation with viral loads at one-year post infection timepoint (shown in orange). **F:** Autologous Tat-mediated T/F LTR transcriptional activity exhibit significant positive correlation with viral loads at one-year post infection timepoint (shown in green). **G:** Consensus Tat-mediated T/F LTR transcriptional activity exhibit significant positive correlation with viral load set point (shown in orange). **H:** Autologous Tat-mediated T/F LTR transcriptional activity exhibit significant positive correlation with viral load set point (shown in green).* One of the ten patients without the matching plasma sample at approximately one-year post infection, had longitudinal plasma viral load measurements thus making a total of 32 patients whose viral load set point could be calculated.

(S4E-S4F Fig). Moreover, transcriptional activity of the T/F LTR sequences exhibiting Sp1III_5A was not significantly different from those with Sp1III_5T (S4G Fig). Viral loads and CD4 T cell counts were not different between participants infected with T/F variants exhibiting Sp1III_5A versus those exhibiting Sp1III_5T variation (S4H-S4I Fig). Taken together these data suggest that HIV-1C T/F LTR sequences exhibit inter-patient genetic variation that translates to differential transcriptional activity and disease outcome.

### *Ex vivo* response of T/F LTR variants to extracellular stimulation

One approach being explored for the development of HIV-1 cure is the "shock and kill" strategy, which uses latency reversing agents (LRAs) to reactivate HIV-1 transcription, protein expression and virion production from latent proviruses [36]. However, previous studies reported that LRAs such as histone deacetylase inhibitors (HDACi) differentially reactivate the virus from latently infected cells [37–39]. Furthermore, variable response to LRAs was also reported in PLWH on suppressive cART [40,41]. Factors that could contribute to this varied response include cellular and viral factors. In this study, we hypothesized that the inter-patient T/F LTR genetic variation and functional differences may be responsible for differential HIV-1 response to extracellular stimulation with LRAs such as PMA, TNF-$\alpha$, Prostratin and SAHA. To address this hypothesis, the response of T/F LTR variants to extracellular stimulation was assessed in the absence or presence of Tat.

Basal T/F LTR transcription activity was significantly enhanced in driving the expression of the *luciferase* gene following extracellular stimulation with PMA ($p < 0.0001$), TNF-$\alpha$ ($p < 0.0001$) and SAHA ($p = 0.0227$), however, this was not the case for Prostratin, possibly due to the absence of Tat (Fig 5A). As expected, Tat-mediated T/F LTR transcriptional activity was significantly higher ($p < 0.001$) following stimulation with PMA, TNF-$\alpha$, Prostratin and SAHA ($p = 0.0002$) compared to Tat-mediated transcriptional activity without stimulation (Fig 5B). These data suggest that the heterogeneous response to LRAs could at least partially be attributed to T/F LTR genetic variation. Significant positive correlation was observed between PMA and Prostratin ($r = 0.4546$; $p = 0.0017$) (Fig 5C) and SAHA ($r = 0.6282$; $p < 0.0001$) (Fig 5D); as well as between Prostratin and SAHA ($r = 0.7840$; $p < 0.0001$) Fig 5E). These data suggest that PMA, Prostratin and SAHA have similar or related mechanisms of viral activation. However, there was no correlation between PMA and TNF-α as well as between TNF-α, Prostratin and SAHA (Fig 5F-5H) suggesting different mechanisms activating of viral gene transcriptions by these stimulants.

## Discussion

Inter- and intra-subtype LTR sequence differences observed during chronic phase of infection may result in differences in functional activity and disease outcome [25,28,42]. Over the course of infection, HIV-1 mutates and diversifies into multiple quasispecies [43]. Although diverse HIV-1 variants are present during the chronic phase of infection, only a single or very limited numbers of variants, termed T/F viruses, establish new infection during heterosexual transmission [3,44,45]. Identifying the phenotypic characteristics of these T/F viruses is important to understand viral factors modulating disease outcome. In this study, we characterized the genetic variation of HIV-1C T/F LTR, and its impact on disease outcome and transcriptional activation potential. LTR sequences generated from plasma obtained during acute HIV-1 infection (Fiebig stages I to VI) and at one-year post infection were sequenced and analyzed for phylogenetic relatedness and other genetic features. Phylogenetic analysis indicated that the HIV-1C T/F LTR sequences exhibited 5.7% (Range: 2–12) inter-patient diversity, with subsequent intrapatient sequence evolution by 12 months post infection observed in 48.4% of the paired samples. The T/F LTR genetic variation translated to differential functional activity and as expected, there was significantly higher Tat-mediated transcriptional activity compared to basal activity. Heterogeneity observed in autologous Tat-mediated transcription could be attributed to genetic variation in autologous Tat, which is 83% (Range 76–92) similar to the consensus, that adds another layer of transcriptional regulation as previously noted [46,47]. However, differential T/F LTR transcription activity in the presence of the consensus Tat could only be attributed to genetic variation within the T/F LTR sequence.

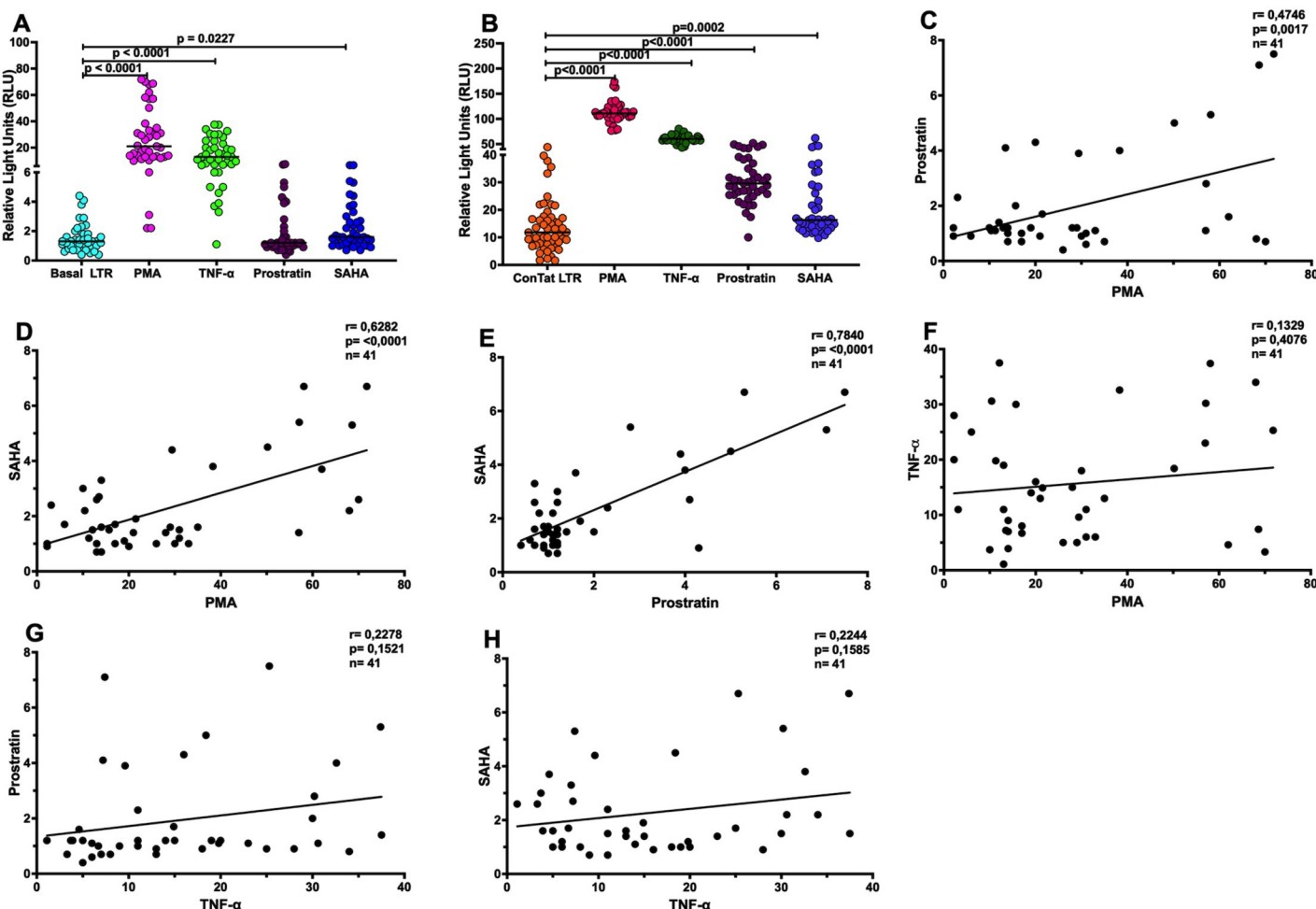

**Fig 5. Interpatient LTR genetic variability translates to differential transcription activity in the presence of cell stimulants.** Patient derived LTR-pGL3 basic vector recombinants were transfected into Jurkat cells alone or in combination with Tat followed by stimulation with PMA, TNF-α, Prostratin and SAHA after 12 hours of incubation. **(A)** Patient derived T/F LTR-pGL3 basic vector recombinants were transfected into Jurkat cells alone to measure basal levels of gene transcription (shown in cyan), and in the presence of stimulation with PMA (shown in pink), TNF-α (shown in green), Prostratin (shown in maroon) and SAHA (shown in blue). **(B)** The patient derived LTR-pGL3 basic vector recombinants were co-transfected with the consensus *tat* into Jurkat cells to measure Tat induced levels of gene transcription (shown in orange) in the absence or presense of stimulation with PMA (shown in cerise), TNF-α (shown in green), Prostratin (shown in plum) and SAHA (shown in violet). **(C)** Significant positive correlation between PMA and Prostratin. **(D)** Positive correlation between PMA and SAHA. **(E)** Significant positive correlation between Prostratin and SAHA. **(F)** No correlation between PMA and TNF-α. **(G)** No correlation TNF-α and Prostratin. **(H)** No correlation between TNF-α and SAHA.

Our data show a significant positive correlation of the basal or Tat-mediated T/F LTR transcriptional activity with viral load but negative correlation with CD4 T cell count at acute infection timepoint. Similarly, both consensus and autologous Tat-mediated transcriptional activity of the T/F LTR variants show a significant positive correlation with viral load but negative correlation with CD4 T cell count at one-year post infection. The viral load set point is a prognostic marker for the rate of disease progression and is a risk factor for sexual transmission [48,49], and therefore our data suggest that T/F virus virulence is at least partially modulated by the viral LTR promoter. In addition to facilitating viral gene transcription by binding to the TAR element on the nascent viral pre-messenger RNA LTR, Tat targets immune cells to promote viral replication and rewire cellular pathways beneficial for the virus thus making the cells more susceptible to infection [50]. Taken together our data suggest that T/F LTR

sequence variation may impact transcription activity and disease outcome. Specifically, the data indicate that LTR-mediated transcriptional activity is an important determinant of clinical disease outcome.

There is genetic variation within the LTR that has significant functional implications [25,26]. Standard HIV-1C viruses have three NF-*k*B binding sites, though some subtype C viral strains include a fourth NF-*k*B binding site [29,30,34,51] that has been reported to enhance the viral promoter at the transcriptional level such that viruses containing the 4 NF-*k*B site dominate the 3 NF-*k*B isogeneic viral strains in pairwise cellular competition assays [28]. Furthermore, mean plasma viral loads were shown to be significantly higher in 4 NF-*k*B HIV-1 infection suggesting that these emergent strains are probably more infectious [28]. However, our study did not show that viruses with 4 NF-*k*B binding sites are being preferentially transmitted or have superior transcriptional potential. This lack of association with functional or clinical outcomes may have been influenced by a limited dataset since there were only three T/F LTR sequences exhibiting the intact fourth NF-*k*B binding site.

Lastly, our data show that basal transcription activity of T/F LTR variants is significantly enhanced following stimulation with different immune activators that have the potential to reverse latency, including PMA, TNF-α, and SAHA. Specifically, PMA showed the strongest activation of T/F LTR variants, mediating *luciferase* gene transcription followed by TNF-α, SAHA, with no difference observed in the presence of Prostratin in the basal condition. Furthermore, as expected, our data showed a 5-fold increase in activation of transcription activity after stimulation with these cell activators in the presence of Tat. In both the absence (basal) or presence of Tat, the expression of the *luciferase* gene after stimulation with cell activators was heterologous, suggesting that sensitivity of T/F LTR variants to activation is variable and that the LTR may be an important determinant of latency reversal following cellular activation. The inability of Prostratin to activate basal transcriptional activity may be attributed to the absence of Tat, since activation of T/F LTR transcription mirrored that of TNF-α in the presence of Tat. Although LRAs including HDACis were shown to activate patient-derived HIV-1B LTRs to similar levels seen with HIV-1 subtype B molecular clone (NL4-3) [52], other studies have reported heterogeneous reactivation of latent viruses [40,41,52,53].

The limitations of this study include that transcriptional assays were perfomed in Jakurt cell line instead of primary cells and may therefore not reflect *in vivo* patterns on virus transcription. We only used one cell line in this study; in the future other cell lines ought to be tested to exclude cell line dependent effects. LTR sequences were analyzed after cloning in an expression plasmid independent of other viral genes, which may limit complex interactions between different regions of the viral genome. Moreover, we did not assess the impact of the T/F viruses LTR variants on the propensity of latency development and/or reactivation.

## Conclusion

Our data demonstrate that HIV-1C T/F LTR sequences vary between PLWH, with subsequent genetic evolution in a substantial proportion of PLWH at one-year post-infection. Furthermore, in this study we show that HIV-1C T/F LTR sequence variation impacts disease outcome and transcriptional activation potential. We observed a low frequency of HIV-1C T/F LTR sequences included the fourth NF-κB binding site, suggesting that the 4 NF-κB viruses are not preferentially transmitted or expanded within the clade-C epidemic in South Africa. Duplication of the NF-κB binding motif and variation within the Sp1III binding site in the core-promoter together with other polymorphisms was not found to be associated with T/F virus LTR transcriptional activity, viral loads or viral loads set point, suggesting complex

mutational patterns that may impact these traits. Future studies should investigate the effect of T/F viruses LTR on latency development or reactivation.

## Materials and methods

### Ethics statement

The study was approved by the Biomedical Research Ethics Committee (BREC) of the University of KwaZulu-Natal (BREC/00001051/2020). All study participants provided written informed consent for participation in the study.

### Study design and participants

A total of 41 PLWH from two acute HIV-1 infection cohorts, the Females Rising through Education, Support and Health (FRESH) study and the HIV Pathogenesis Programme (HPP) acute infection (AI) cohort were included in this study. FRESH is an ongoing prospective cohort, established in Durban, KwaZulu-Natal, South Africa in 2012 to study hyperacute HIV-1 infection as previously described [54–56]. Briefly, young women between 18–23 years old who are not living with but at high risk of HIV infection, are enrolled in a 9-month study that is integrated with a lifeskills and empowerment programme that includes intensive HIV prevention counseling and provision of oral pre-exposure prophylaxis (PrEP). HIV-1 RNA testing is performed twice per week to coincide with the empowerment classes schedule, and enabling detection of acute HIV infection during Fiebig stage I infection [Ref 57]. To date, FRESH has enrolled 2,723 women and identified 98 with acute HIV-1 infections. Acute HIV-1 infection staging was based on the classification of Fiebig et al. [57], including plasma HIV-1 RNA, plasma p24 antigen (p24 Ag), Western blot (WB), though used fourth generation HIV enzyme immunoassay (EIA). In accordance with South African national treatment guidelines at the time, the first 14 individuals diagnosed with acute infection were not started on antiretroviral therapy immediately and therefore remained treatment-naïve for a median period of 344 days. Plasma samples from these 14 individuals and an additional 7 acutely infected women from the FRESH cohort who started antiretroviral therapy at 1 to 3 days after HIV-1 RNA detection were analyzed at the earliest post infection timepoint available (pre-ART initiation). Matching plasma samples obtained one-year post-infection were also analyzed for 14 FRESH patients who remained antiretroviral-naïve.

The HIV Pathogenesis Programme (HPP) Acute Infection cohort was also based in Durban, KwaZulu-Natal, South Africa, and it enrolled men and women aged 18–24 years with acute HIV infection as previously described [58–60]. Briefly, individuals testing HIV negative by rapid immunoassays (SD Bioline HIV1/2 Elisa 3.0, Standard Diagnostics Inc, Kyonggi-do, Korea [3rd generation] and Vironostika HIV-1 Uni-Form II Plus O v5.0 Biomérieux, Durham, North Carolina, USA) at routine testing sites were consented to undergo HIV-1 RNA testing. Individuals with a positive HIV-1 RNA test and a negative HIV-1 ELISA test (at the time of first RNA positive test) and an evolving Western blot (GS HIV Type 1 Western blot, Bio-Rad, Redmond, WA, USA) pattern indicative of a recent HIV-1 infection were enrolled. For this cohort, the date of infection was estimated to be 14 days prior to the first positive HIV-1 RNA as previously described [7,58,60]. Blood samples were collected at enrollment, 2 weeks, 4 weeks, 2-, 3- and 6-months post HIV-1 RNA detection and then every 6 months thereafter. Viral loads were measured at all study visits using the Roche Cobas Taqman HIV-1 Test v2.0 (Roche Diagnostics, Branchburg, NJ, USA). CD4 T cell counts were also enumerated at all study visits using the 4-colour MultiTEST/Trucount assay (Becton Dickinson, San Jose, CA, USA) and analyzed further by flow cytometry on a FACSCalibur (BD Biosciences, San Jose, CA, USA). The median treatment-free follow-up time for the patients was 376 days [IQR,

354–430] post infection. Plasma samples from 20 PLWH at the earliest post infection time-point and 17 matching plasma samples at one-year post infection timepoint were analyzed.

## Viral RNA extraction, HIV-1 LTR U3R region amplification, sequencing and data analysis

Viral RNA was extracted from 140 μL of plasma using the QIAamp Viral RNA Extraction Mini Kit (Qiagen, Hilden, Germany) as per the manufacturer's instructions. Reverse transcription and first round polymerase chain reaction (PCR) were performed using a SuperScript™ III One-Step RT-PCR System with Platinum™ *Taq* DNA Polymerase kit (Invitrogen, Carlsbad, CA, USA). Briefly, each 25 μL RT-PCR reaction mixture constituted of 2 μL of extracted viral RNA, 1X Ready Reaction Mix, 0.2 μM of reverse primer 3' R-M: 5'
ACTYAAGGCAAGCTTTATTGAG 3' (HXB2 nucleotide [nt] 9630 to 9609) and forward primer U3-M: 5' CAGGTACCTTTAAGACC–AATGAC 3' (nucleotide [nt] 9013 to 9035) [61], 1 U SuperScript™ III RT/Platinum™ TaqMix and this was made up to the total reaction with diethyl pyrocarbonate (DEPC)-treated water. Thermocycler conditions were as follows: 45°C for 30 min and 94°C for 15 s followed by 40 cycles of 94°C for 15 s, 55°C for 30 s, 68°C for 2 min and 68°C for 7 min. The nested PCR was performed to amplify the U3R region of the viral LTR using the inner primers, forward primer 5'T7-LTRKpnI (5'
*taatacgactcactataggg*TT GGTACC TTTAAAAGAAAAGGGGGGAC 3', nt 9064 to 9085, T7 primer sequence in *italics* and *Kpn*I-site underlined) and reverse primer 3'Sp6–LTRHin-dIII: 5' *atttaggtgacactatag*ATTGAGG AAGCTT CAGTGGG 3' (nt 9614 to 9593, Sp6 primer sequence in *italics* and *Hind*III underlined) adapted from a previous study [61], using KAPA HiFi HotStart DNA Polymerase PCR Kit KAPA Biosystems (Cape Town, South Africa). Briefly, a 50 μL PCR mixture was prepared for each sample, comprising 1 X KAPA HiFi Buffer, 0.2 mM of deoxynucleoside triphosphates (dNTPs), 0.2 μM of each primer, 1 μL RT-PCR product and 1 U KAPA HiFi HotStart DNA Polymerase and DEPC water was used to bring the mixture to the final reaction volume. Thermocycler conditions were as follows: 95°C for 5 min; 25 cycles of 98°C for 20 s, 55°C for 15 s, and 72°C for 15 sec; and 72°C for 5 min. The PCR products were analyzed on 1% agarose gel and the correct size DNA band was gel extracted using GeneJet Gel Extraction Kit (ThermoScientific, Waltham, MA, USA) as per the manufacturer's instruction. The rest of gel extracted DNA was put aside for cloning while 20 ng DNA was used for sequencing using the Big Dye Terminator v3.1 Cycle Sequencing Reaction (Applied Biosystems, California, USA). Sequencing samples were analyzed using the ABI 3130xl Genetic Analyzer.

The HIV-1 LTR U3R sequences were assembled and edited using the Sequencher Program v5.0 (Gene Codes Corporation, Ann Arbor, MI, USA). Phylogenetic relatedness to compare and evaluate inter- and intra-patient diversity was performed by Neighbor-Joining trees (with 1,000 bootstrap replicates) using PhyML Maximum Likelihood software (https://www.hiv.lanl.gov/content/sequence/PHYML/interface.html). Branching topology was visualized in Figtree (http://tree.bio.ed.ac.uk/software/figtree). Multiple sequence alignment was done using MAFFT and visualized using BioEdit. The HIV-1C reference strain, EF178613.1 was obtained from the Los Alamos HIV sequence database (www.hiv.lanl.gov).

## Genetic distance

We examined consensus-level SNP distances between interpatient sequences at the early time point and intrapatient sequences between the early and the late time points pairs using the snp-dists package. Sequences with SNP distance between 0 and 2 were considered highly similar.

## Generation of the LTR U3R-pGL3 luciferase reporter constructs

The HIV-1C consensus LTR sequence (amplified from the following reagent obtained through the NIH HIV Reagent Program, Division of AIDS, NIAID, NIH: Human Immunodeficiency Virus-1 (HIV-1) 93ZM74 LTR Luciferase Reporter Vector, ARP-4789, contributed by Dr. Reink Jeeninga and Dr. Ben Berkhout) or remainder of gel extracted DNA amplified from LTR U3R of the 41 RNA samples obtained at the earliest post infection timepoint (referred to as T/F LTR) were cloned into the pGL3 Basic vector expressing Firefly Luciferase (Promega Corporation, Wisconsin, USA) as previously described [62]. Briefly, the pGL3 Basic vector and consensus or T/F LTR U3R PCR products were digested with *Kpn*I and *Hind*III (New England Biolabs, Ipswich, MA, USA), individually as per manufacturer's instruction. The restriction fragments were analyzed on a 1% agarose gel, correct size fragments corresponding to the consensus or T/F LTR U3R regions and linearized pGL3 Basic vector were gel extracted using the GeneJet Gel extraction kit (ThermoFisher Scientific, Waltham, MA, USA) as per the manufacturer's instructions. The digested consensus or T/F LTR U3R region was cloned into the linearized pGL3 Basic vector/plasmid and ligated using 1 U of T4 DNA ligase (New England Biolabs, MA, USA) as per manufacturer's instructions, to generate recombinant consensus or T/F LTR U3R-pGL3 *luciferase* reporter plasmid. The recombinant consensus or T/F LTR U3R-pGL3 *luciferase* reporter plasmids were transformed into JM109 competent *E. coli* cells (Promega, USA, Madison) as per manufacturer's instructions and grown overnight on ampicillin agar plates at 37˚C for 14 hours. The recombinant consensus or T/F LTR U3R-pGL3 reporter plasmid was then purified from a randomly picked colony using GeneJet Plasmid Mini Prep Kit (Invitrogen, Carlsbad, CA) as per the manufacturers' instructions. The insert, consensus or T/F LTR U3R region was confirmed by sequencing as aforementioned. Subsequently, the large stocks of recombinant consensus or T/F LTR-pGL3 reporter plasmids were generated using Plasmid Maxi kit (Qiagen, Valencia, CA) for immediate use or longer storage at -80˚C.

## Analysis of the Reporter Gene Expression

Jurkat cells were transiently transfected with the Polyethylenimine (PEI) transfection reagent (ThermoScientific, Waltham, MA, USA) as previously described [28,63]. Briefly, Jurkat cells were seeded into 24-well tissue culture plates at a density of $5 \times 10^5$ cells/well in 400 μL of antibiotic-free RPMI 1640 medium supplemented with 10% FBS. A plasmid DNA pool of 400 ng, containing 300 ng of the recombinant T/F LTR-pGL3 plasmid and 100 ng of recombinant control insert DNA-pTarget plasmid (Promega Corporation, Madison, WI) for the Tat minus transfection or 100 ng of recombinant HIV-1C consensus (amplified from the following reagent obtained through the NIH HIV Reagent Program, Division of AIDS, NIAID, NIH: Plasmid pcDNA3.1(+) Expressing Isogenic Mutant Human Immunodeficiency Virus Type 1 Subtype C BL43/02 Tat (pC-Tat.BL43.CC), ARP-11785, contributed by Dr. Udaykumar Ranga) or autologous Tat (sequences amplified from the same sample from which the LTR was derived (S1A Fig) with an average of 83% (Range: 76–92) (S1B Fig)) cloned into pTarget plasmid was prepared in 50 μL of serum-free RPMI medium. Briefly, 1 μL of PEI was mixed with 49 μL of serum-free RPMI medium to prepare the lipid transfection reagent. Then, 50 μL PEI-RPMI mixture was mixed with 50 μL of the recombinant T/F LTR-pGL3 plasmid and recombinant control insert DNA-pTarget or recombinant consensus or autologous Tat-pTarget plasmids. The 100 μL plasmid-lipid mixture was then incubated for 20 minutes at room temperature and subsequently added to the corresponding wells of a 24-well plate. Twelve hours following the transfection, the cells were washed to remove the lipid complexes and resuspended in 500 μL of R10 medium. The transfection reaction was then incubated for 24

hours followed by a luciferase assay. Transiently transfected Jurkat cells in a volume of 500 μL were incubated in the absence or presence of the activators: TNF-α (20 ng/mL; R&D Systems, Minneapolis, MN), Phorbol 12-myristate 13-acetate (PMA, 20 ng/mL, Sigma-Aldrich, St. Louis, Missouri, USA), Prostratin (20 ng/mL; Sigma-Aldrich, St. Louis, Missouri, USA) or SAHA (20 ng/mL; Sigma-Aldrich, St. Louis, Missouri, USA M) as previously described [35]. Bright-Glo (Promega, Madison, USA) was used to monitor the levels of the Luciferase expression at 24 hours. The Luciferase assay was performed using the Victor Nivo Multimode plate recorder (PerkinElmer, Massachusetts, USA) by mixing 150 μL of cell culture and 100 μL of Bright-Glo substrate reagent into the wells of a black round bottom 96 well plate. The experiments were performed in triplicate wells, and every experiment was repeated at least two times.

## Statistical analyses

Statistical analysis was performed using Graphpad Prism 5 software (San Diego California USA). The statistical significance for the transfection assay and association of viral load with LTR mutants was determined using a paired T-test. Linear regression analyses were used to determine the statistical significance and correlation coefficient of the correlation between transcription activity and markers of disease progression. A p value <0.05 was considered statistically significant.

## Supporting information

**S1 Fig. Multiple sequence alignment of patient derived Tat sequences and percentage of similarity between patient derived and consensus Tat sequences. A:** The patient derived Tat sequences were aligned against the Zambian subtype C consensus Tat sequence, amplified from the following reagent that was obtained through the NIH HIV Reagent Program, Division of AIDS, NIAID, NIH: Plasmid pcDNA3.1(+) Expressing Isogenic Mutant Human Immunodeficiency Virus Type 1 Subtype C BL43/02 Tat (pC-Tat.BL43.CC), ARP-11785, contributed by Dr. Udaykumar Ranga. **B:** Shows the percentage of similarity of each individual patients derived LTR at early time point (ETP) to the consensus sequence.
(TIFF)

**S2 Fig. Multiple sequence alignment of patient derived LTR sequences exhibiting the TAR element.** The patient derived TAR element sequences were aligned against the Indian subtype C reference sequence, EF178613.1. The dots represent nucleotide bases that are identical to the reference sequence. The doted block highlight the TAR element, TAR loop region. Overall, the T/F virus canonical sequence of the TAR loop region was relatively conserved.
(TIFF)

**S3 Fig. Correlation of T/F LTR transcriptional activity with markers of disease progression. A:** Basal T/F LTR transcriptional activity exhibit significant negative correlation with CD4 T cell counts at or near transmission (shown in Cyan). **B:** Consensus Tat-mediated T/F LTR transcriptional activity exhibit significant negative correlation with CD4 T cell counts at or near transmission (shown in orange). **C:** Autologous Tat-mediated T/F LTR transcriptional activity exhibit significant negative correlation with CD4 T cell counts at or near transmission (shown in green). **D:** Basal T/F LTR transcriptional activity shows no relationship with viral loads at one-year post infection timepoint (shown in Cyan). **E:** Basal T/F LTR transcriptional activity shows no relationship with CD4 T cell counts at one-year post infection (shown in Cyan). **F:** Consensus Tat-mediated T/F LTR transcriptional activity exhibit significant negative correlation with CD4 T cell counts at one post infection timepoint (shown in orange). **G:**

Autologous Tat-mediated T/F LTR transcriptional activity exhibit significant negative correlation with CD4 T cell counts at one-year timepoint (shown in green). **H:** Basal T/F LTR transcriptional activity shows no relationship with viral load set point (shown in Cyan). * One of the ten patients without the matching plasma sample at approximately one-year post infection, had longitudinal plasma viral load measurements thus making a total of 32 patients whose viral load set point could be calculated.
(TIFF)

**S4 Fig. Association of genetic variation within transcription factor binding sites (TFBS) with transcription activity and markers of disease progression.** Each point in the plot represents an HIV-1 positive individual. **A-C**: No difference in transcriptional activity between the T/F LTR sequences exhibiting 3 NF-$k$B (shown in blue) versus 4 NF-$k$B (shown in tangerine) similarly viral loads and CD4 T cell counts were not different between patients infected with T/F viruses exhibiting 3 NF-$k$B sites versus 4 NF-$k$B sites. **D-F**: Transcriptional activity between the T/F LTR sequences exhibiting Sp1III_2G (shown in mocha) versus Sp1III_2A (shown in grey) were not different and similarly viral loads and CD4 T cell counts were not different between patients infected with T/F viruses Sp1III_2G versus Sp1III_2A. **G-I**: Transcriptional activity between the T/F LTR sequences exhibiting Sp1III_5A (shown in yellow) versus Sp1III_5T (shown in navy blue) were not different and similarly viral loads and CD4 T cell counts were not different between patients infected with T/F viruses Sp1III_5A versus Sp1III_5T.
(TIFF)

**S1 Table. The demographic and clinical characteristics showing absolute CD4 T cell count of the 41 people living with HIV-1 used in this study.**
(DOCX)

## Acknowledgments

The authors would like to acknowledge the study participants and the clinical teams from the FRESH and HIV Pathogenesis Programme (HPP) acute infection studies.

## Author Contributions

**Conceptualization:** Paradise Madlala, Thumbi Ndung'u.

**Data curation:** Paradise Madlala, Zakithi Mkhize, Shamara Naicker, Samukelisiwe P. Khathi, Krista L. Dong.

**Formal analysis:** Paradise Madlala, Samukelisiwe P. Khathi, Shreyal Maikoo, Kasmira Gopee.

**Funding acquisition:** Paradise Madlala, Thumbi Ndung'u.

**Methodology:** Paradise Madlala.

**Supervision:** Paradise Madlala.

**Writing – original draft:** Paradise Madlala, Thumbi Ndung'u.

**Writing – review & editing:** Paradise Madlala, Zakithi Mkhize, Shamara Naicker, Samukelisiwe P. Khathi, Shreyal Maikoo, Kasmira Gopee, Krista L. Dong, Thumbi Ndung'u.

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
