## [Decision Letter · Decision Letter 0]

23 Mar 2023

Dear Dr Madlala,

Thank you very much for submitting your manuscript "Genetic variation of the HIV-1 subtype C transmitted/founder viruses long terminal repeat elements and the impact on transcription activation potential and clinical disease outcomes" for consideration at PLOS Pathogens. As with all papers reviewed by the journal, your manuscript was reviewed by members of the editorial board and by several independent reviewers. The reviewers appreciated the attention to an important topic. Based on the reviews, we are likely to accept this manuscript for publication, providing that you modify the manuscript according to the review recommendations.

Studies on genetic diversity of the LTR of HIV subtype C prevailing in South Africa are of interest in the context of HIV latency, persistence.

The paper was now seen by 1 external referee and a guest editor.

This longitudinal analysis seems well performed. Still, authors should take the comments and suggestions as listed below into consideration.

Sincerely,

Zeger Debyser

Guest Editor

PLOS Pathogens

Susan Ross

Section Editor

PLOS Pathogens

Kasturi Haldar

Editor-in-Chief

PLOS Pathogens

orcid.org/0000-0001-5065-158X

Michael Malim

Editor-in-Chief

PLOS Pathogens

orcid.org/0000-0002-7699-2064

Studies on genetic diversity of the LTR of HIV subtype C prevailing in South Africa are of interest in the context of HIV latency, persistence.

The paper was now seen by 1 external referee and a guest editor.

This longitudinal analysis seems well performed. Still, authors should take the comments and suggestions as listed below into consideration.

Reviewer Comments (if any, and for reference):

Reviewer's Responses to Questions

**Part I - Summary**

Reviewer #1: This a longitudinal analysis of HIV subtype C LTR diversity and transcriptional activation during acute / early infection in South Africa. The patient population, in vitro methods ,and data analysis are well described. Overall, this is a very nice study with interesting findings.

Reviewer #2:

Stengths

Detailed characterisation of the genetic evolution of the HIV LTR from subtype C strains in the first year after acute infection in 41 patients with a focus on transcriptional activity. Numbers on genetic diversity and evolution are given. A correlation between transcriptional activity and viral loads is demonstrated. Heterogeneous response to LRA activation is shown. No evidence for selection of 4 NF-kappaB sites in the LTR in South African population in contrast to previous reports from India.

Strengths are the focus on epidemiologically relevant subtype C strains and the data collection.

Weakness

Not much mechanistic insight is provided.

**Part II – Major Issues: Key Experiments Required for Acceptance**

Reviewer #1: N/A

Reviewer #2: No additional experiments required.

**Part III – Minor Issues: Editorial and Data Presentation Modifications**

Reviewer #1: A few minor revisions would strengthen this manuscript further, including:

• Lines 145 and 148 and Table 1: Why is the median square root CD4 T cell count presented rather than the absolute value?

• Line 178: what is the range of pairwise diversity between the early and late time points? Were there any paired samples that showed no / very little inter-time point diversity?

• Additional details on consensus versus autologous Tat sequences are needed. Were autologous Tat sequences derived from a previous study? How similar / different are they from the consensus Tat sequence?

• The diversity inherent in Figure 2 is difficult to see. A phylogram may be better than a circular tree for data visualization / presentation.

• The use of Jurkat cells is appropriate. However, showing some results in another cell type would be helpful for confirming that results are not cell type or cell line dependent.

Reviewer #2:

Please rephrase line 175-177 on page 6. Not clear what authors want to tell.

Line 360 : perhaps a number is missing (4 NF-kappaB sites)

PLOS authors have the option to publish the peer review history of their article (what does this mean?). If published, this will include your full peer review and any attached files.

Reviewer #1: No

Reviewer #2: No

Figure Files:

Data Requirements:

Reproducibility:

References:

---

## [Decision Letter · Decision Letter 1]

4 May 2023

Dear Dr. Madlala

We are pleased to inform you that your manuscript 'Genetic variation of the HIV-1 subtype C transmitted/founder viruses long terminal repeat elements and the impact on transcription activation potential and clinical disease outcomes' has been provisionally accepted for publication in PLOS Pathogens.

Please address the 3 minor editorial comments in the final manuscript.

Best regards,

Zeger Debyser

Guest Editor

PLOS Pathogens

Susan Ross

Section Editor

PLOS Pathogens

Kasturi Haldar

Editor-in-Chief

PLOS Pathogens

orcid.org/0000-0001-5065-158X

Michael Malim

Editor-in-Chief

PLOS Pathogens

orcid.org/0000-0002-7699-2064

Editorial comments:

1. To this effect, a new sentence has been added on page 19 lines 383 – 385

“While only one cell line was used in this study, it is possible that the use of other cell lines may have yielded same results thus confirming that these results are not cell type dependent or shown different results.”

Replace this sentence by

“We only used one cell line in this study; in the future other cell lines ought to be tested to exclude cell line dependent effects.”

2. The inter timepoint genetic distance data includes a lower value (range) that is negative. This is not possible! the lowest value that genetic distance can have is 0.0% / no change. Please correct.

3. last sentence of abstract, typo : potential instead of potentional

Reviewer Comments (if any, and for reference):

Reviewer's Responses to Questions

**Part I - Summary**

Reviewer #1: (No Response)

**Part II – Major Issues: Key Experiments Required for Acceptance**

Reviewer #1: (No Response)

**Part III – Minor Issues: Editorial and Data Presentation Modifications**

Reviewer #1: The inter timepoint genetic distance data includes a lower value (range) that is negative. This is not possible! the lowest value that genetic distance can have is 0.0% / no change.

PLOS authors have the option to publish the peer review history of their article (what does this mean?). If published, this will include your full peer review and any attached files.

Reviewer #1: No

---

## [Editor Report · Acceptance letter]

8 Jun 2023

Dear Dr Madlala,

We are delighted to inform you that your manuscript, "Genetic variation of the HIV-1 subtype C transmitted/founder viruses long terminal repeat elements and the impact on transcription activation potential and clinical disease outcomes," has been formally accepted for publication in PLOS Pathogens.

Best regards,

Kasturi Haldar

Editor-in-Chief

PLOS Pathogens

orcid.org/0000-0001-5065-158X

Michael Malim

Editor-in-Chief

PLOS Pathogens

orcid.org/0000-0002-7699-2064